# Backbone amides are determinants of Cl⁻ selectivity in CLC ion channels

Lilia Leisle[1], Kin Lam[2,3,4,5,6], Sepehr Dehghani-Ghahnaviyeh [2,3,4,5,6], Eva Fortea[1,7], Jason D. Galpin [8], Christopher A. Ahern [8], Emad Tajkhorshid [2,3,4,5,6] & Alessio Accardi [1,7,9] ✉

Chloride homeostasis is regulated in all cellular compartments. CLC-type channels selectively transport Cl⁻ across biological membranes. It is proposed that side-chains of pore-lining residues determine Cl⁻ selectivity in CLC-type channels, but their spatial orientation and contributions to selectivity are not conserved. This suggests a possible role for mainchain amides in selectivity. We use nonsense suppression to insert α-hydroxy acids at pore-lining positions in two CLC-type channels, CLC-0 and bCLC-k, thus exchanging peptide-bond amides with ester-bond oxygens which are incapable of hydrogen-bonding. Backbone substitutions functionally degrade inter-anion discrimination in a site-specific manner. The presence of a pore-occupying glutamate side chain modulates these effects. Molecular dynamics simulations show backbone amides determine ion energetics within the bCLC-k pore and how insertion of an α-hydroxy acid alters selectivity. We propose that backbone-ion interactions are determinants of Cl⁻ specificity in CLC channels in a mechanism reminiscent of that described for K⁺ channels.

Anion-selective channels and transporters control Cl⁻ homeostasis in all living cells and within their intracellular compartments. The ability of these channels to select against cations and discriminate amongst physiological anions is central to their function in vivo. While cation selectivity mechanisms are relatively well understood[1–8], the principles underlying Cl⁻ channel selectivity are poorly resolved. Indeed, most 'Cl⁻ channels' are more permeable to anions other than their biological namesake: GABA[9], CFTR[10], TMEM16A[11], TMEM16B[12], and Bestrophin[13,14] channels follow the Hofmeister lyotropic selectivity sequence[15] of SCN⁻>I⁻>NO₃⁻>Br⁻>Cl⁻, with slight deviations. In contrast, CLC-type channels and transporters select for Cl⁻ over other anions with a sequence of Cl⁻>Br⁻>NO₃⁻>I⁻[16–21]. This selectivity sequence is evolutionarily well-conserved from prokaryotes to

eukaryotes and between transporters and channels. While most CLCs are Cl⁻ selective, the atCLC-a exchanger from *Arabidopsis thaliana* is NO₃⁻ selective[22,23], and members of a clade of prokaryotic CLCs are highly F⁻ selective[24–27]. Thus, the CLC pore provides a unique and plastic structural template to investigate the mechanisms that underlie anion selectivity.

All CLCs are dimers, where each monomer forms a separate Cl⁻ permeation pathway[28,29]. The CLC-ec1 structure allowed for the identification of three anionic binding sites[30–37], coined $S_{int}$, $S_{cen}$ and $S_{ext}$ for internal, central and external, respectively (Fig. 1A–C), whose position and coordination are evolutionarily conserved in eukaryotic channels and transporters. Coordination with permeant anions at the internal $S_{int}$ site is weak as the dehydration of ions is

[1]Department of Anesthesiology, Weill Cornell Medical College, New York, NY, USA. [2]Theoretical and Computational Biophysics Group, University of Illinois at Urbana-Champaign, Urbana, IL 61801, USA. [3]NIH Center for Macromolecular Modeling and Bioinformatics, University of Illinois at Urbana-Champaign, Urbana, IL 61801, USA. [4]Beckman Institute for Advanced Science and Technology, University of Illinois at Urbana-Champaign, Urbana, IL 61801, USA. [5]Department of Biochemistry, University of Illinois at Urbana-Champaign, Urbana, IL 61801, USA. [6]Center for Biophysics and Quantitative Biology, University of Illinois at Urbana-Champaign, Urbana, IL 61801, USA. [7]Department of Physiology and Biophysics, Weill Cornell Medical College, New York, NY, USA. [8]Department of Molecular Physiology and Biophysics, University of Iowa, Iowa City, IA, USA. [9]Department of Biochemistry, Weill Cornell Medical College, New York, NY, USA. ✉e-mail: ala2022@med.cornell.edu

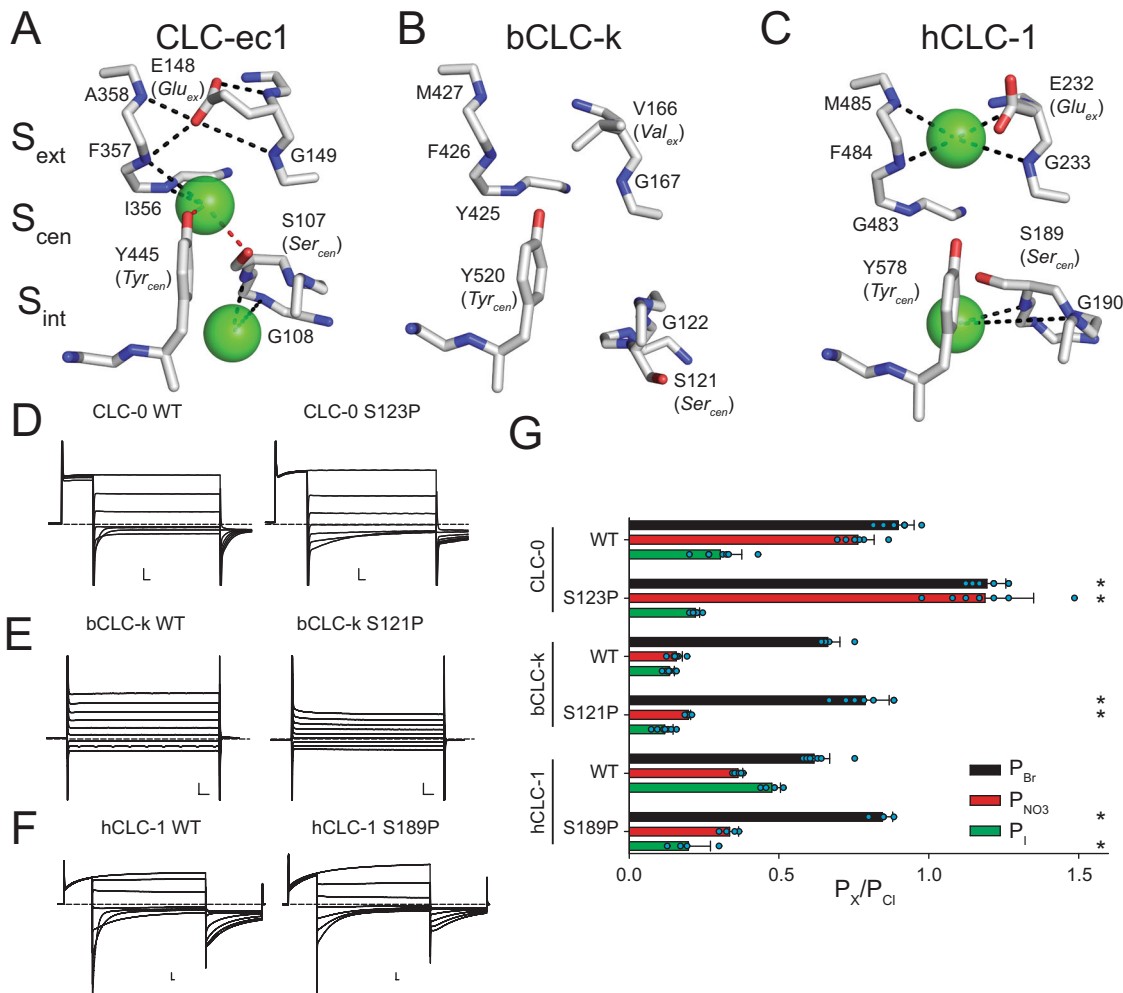

**Fig. 1 | Structural architecture and ion coordination in the Cl⁻ pathway of CLC channels and transporters.** Close up view of the Cl⁻ permeation pathway in CLC-ec1 (PDB: 1OTS, **A**), bCLC-k (PBD: 5TQQ, **B**) and CLC-1 (PDB: 6COY, **C**). The position of the external ($S_{ext}$), central ($S_{cen}$) and internal ($S_{int}$) binding sites is identified based on the crystal structure of CLC-ec1[31]. Bound Cl⁻ ions are shown as green spheres. No ions were resolved in the bCLC-k structure[33]. Dashed lines indicate hydrogen bonds between the Cl⁻ ions and side chains (red) or backbone amides (black)[31,34]. Representative current traces of WT and proline mutants at $Ser_{cen}$ in CLC-0 (**D**), bCLC-k (**E**) and hCLC-1 (**F**). Dashed lines indicate the 0 current level.

Scale bars indicate 2 µA and 10 ms. **G** The relative permeability ratios for Br⁻, NO₃⁻ and I⁻ of CLC-0 (WT and S123P), bCLC-k (WT and S121P) and hCLC-1 (WT and S189P). Data are Mean ± St.Dev. of $n > 4$ repeats from $N \geq 3$ independent oocyte batches, individual data points are shown as cyan circles. The statistical significance of the effects of the mutants on the permeability ratios of each ion (indicated by *) was evaluated with a one-sided Student's t-test with a Bonferroni correction (see Methods). Mean values and p-values are reported in Supplementary Table 1. Individual data points are shown grouped by ion in Supplementary Fig. 5. Raw data for **D**–**G** and Fig. 2B, C is included in the Source Data Files.

only partially compensated via interactions with the backbone amides of a serine residue at position 107 (called $Ser_{cen}$) and Glycine 108 (using CLC-ec1 numbering) in the loop connecting helices C and D (Fig. 1A, C)[19,30,38]. Conversely, anions positioned in the central and external sites, $S_{cen}$ and $S_{ext}$, interact with the protein more extensively, consistent with a key role of this region in selectivity. In $S_{cen}$, Cl⁻ is coordinated by the conserved side chains of S107 ($Ser_{cen}$) and of Y445 ($Tyr_{cen}$), as well as the backbone amides of I356 and F357 (Fig. 1A). Ion coordination in $S_{ext}$ is mediated by the backbone amides of E148 ($Glu_{ex}$), G149, F357 and A359 (Fig. 1A). Thus, interactions with side chains and backbone amides contribute to the preferential stabilization of anions over cations within the CLC pore[31,39]. This pore architecture is conserved in the mammalian bCLC-k and hCLC-1 channels, involving similar coordination patterns (Fig. 1B, C), with the notable exception that in bCLC-k $Ser_{cen}$ points away from $S_{cen}$ (Fig. 1B). The negatively charged side chain of $Glu_{ex}$ is a tethered anion that can occupy the $S_{cen}$ and $S_{ext}$ sites with similar coordination to that of the bound Cl⁻ ions[31,32,39] (Fig. 1,

Supplementary Fig. 1). The competition between $Glu_{ex}$ and the Cl⁻ ions is essential for CLC function[28,29,32,40–42] and weakened ion binding at the $S_{cen}$ site alters Cl⁻/H⁺ exchange stoichiometry in the transporters and gating in channels[17,20,43–49]. Thus, the molecular and energetic determinants of selective anion binding and permeation also govern the CLC transport mechanism.

The current consensus mechanism for CLC selectivity is that the $S_{cen}$ site is the primary regulator of anion discrimination and that the side chain of $Ser_{cen}$ is the critical determinant of its specificity, as proline mutations at this site switch the selectivity from Cl⁻ to NO₃⁻ and vice versa[17,19–21,23,50–52]. However, the recent finding that in bCLC-k channel $Ser_{cen}$ points away from $S_{cen}$[33] (Fig. 1B) and that mutations at $Ser_{cen}$ in the human CLC-Ka channel do not affect selectivity[53], recently led to the proposal that other pore-lining side chains are important for anion specificity. However, while side chains are not conserved, the functional preservation of the anion selectivity sequence points to a shared mechanism between hCLC-Ka and other CLC channels and transporters.

The extensive hydrogen bonding network of permeating anions with pore-lining backbone amides (Fig. 1A–C) led us to hypothesize that backbone amides might provide the conserved pattern of anion coordination in the CLCs, while side chain interactions contribute to fine-tuning of ion selectivity. Using a combination of atomic mutagenesis, electrophysiology, and molecular dynamics (MD) simulations we show here that anion selectivity in CLC-0 and bCLC-k is determined by pore-lining backbone amides and their replacement with an ester oxygen destabilizes Cl⁻ binding with parallel effects on ion selectivity and permeation. Our results suggest that the role of backbone amides in ion selectivity depends on the side chain at the 'gating glutamate' position. Taken together, our results shed new light onto the mechanism of anion permeation and selectivity in a CLC channel and show that backbone amides are critical in allowing these channels to specifically select Cl⁻ over other anions.

## Results

### C-D loop orientation does not determine the role of $Ser_{cen}$ in anion selectivity

We tested whether the structural arrangement of the C-D loop (Fig. 1A–C) determines the role of $Ser_{cen}$ in CLC selectivity by replacing this residue with a proline in CLC-0, CLC-1 and bCLC-k (Fig. 1D–G, Supplementary Figs. 2, 3, 5, Supplementary Table 1). Consistent with past results, the anion selectivity of CLC-0 is drastically altered by the S123P mutation, with the mutant becoming more permeable to Br⁻ and $NO_3^-$ than Cl⁻ (Fig. 1D, G, Supplementary Fig. 2, 3, 5, Supplementary Table 1)[19,21]. However, absent direct structural information on this channel it is difficult to interpret this effect. Thus, we introduced the corresponding mutation in the structurally known bCLC-k and hCLC-1 channels that differ in the orientation of $Ser_{cen}$ (Fig. 1B, C, E, F, Supplementary Figs. 2, 3, 5, Supplementary Table 1)[33–35]. Unlike S123P CLC-0, mutant constructs of bCLC-k and hCLC-1 retain the selectivity sequences of their parent channels, Cl⁻>Br⁻>$NO_3^-$-I⁻, with small alterations (Fig. 1G). While the lack of effect of the S121P mutant of bCLC-k could be rationalized based on the different orientation of the C-D loop in this channel, the lack of effect of S189P in hCLC-1 is more surprising. The wider intracellular constriction in CLC-1 could weaken binding to $S_{cen}$, as suggested by the lack of density at this site in the cryo-EM structure (Fig. 1C)[34,35]. Alternatively, the C-D loop could interchange between the conformations seen in CLC-1 and bCLC-k, so that the importance of the hydrogen bond between an ion in $S_{cen}$ and $Ser_{cen}$ could depend on the stability of the two states. These results suggest other structural elements might play a more important role in determining the conserved selectivity sequence in CLCs.

### Backbone amides contribute to anion selectivity in bCLC-k and CLC-0

To test the role of backbone amides in anion selectivity we used the nonsense suppression method to site-specifically replace amino acids whose backbone amides may participate in ion coordination with their α-hydroxy acid equivalents[54,55]. This atomic manipulation "mutates" the peptide bond into an ester bond by substituting the backbone NH group with an oxygen atom (Fig. 2A), thus eliminating the backbone's ability to function as an H-bond donor, without altering side chain properties. The introduced ester is an otherwise modest change that shares similar bond lengths, angles, preference for a trans geometry, and comparably high energy barrier for rotation[55,56]. We chose the bCLC-k and CLC-0 channels as representatives CLC channels where $Ser_{cen}$ does not (bCLC-k) or does (CLC-0) control anion selectivity. Incorporation of the α-hydroxy acids at the tested positions in bCLC-k and CLC-0 results in robust currents with measurable shifts in reversal potentials in different anions (Supplementary Figs. 2–4). The ratio of the currents measured in oocytes injected with tRNA conjugated to the UAA or with unconjugated tRNA is >9 at all positions (Supplementary

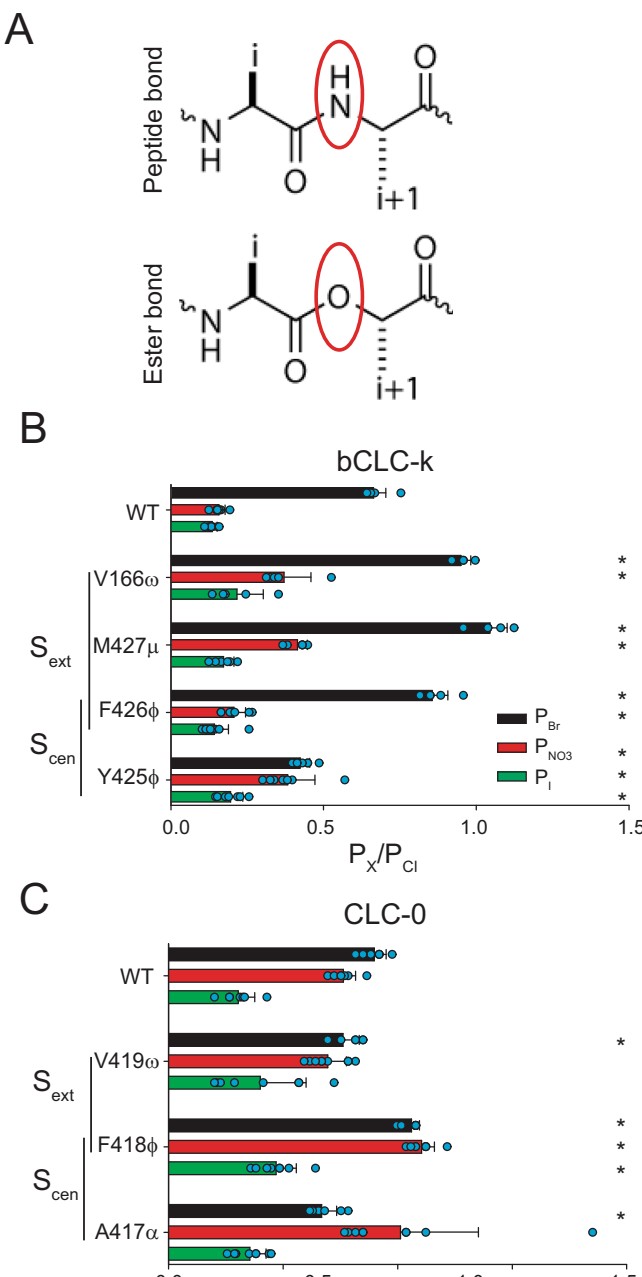

**Fig. 2 | Role of backbone amides in anion selectivity of CLC-0 and bCLC-k.**
**A** Schematic representation of peptide (top panel) and ester bonds (bottom panel). (**B, C**) Effect of replacing backbone amides with ester oxygens at positions lining $S_{ext}$ and $S_{cen}$ in bCLC-k (**B**) and CLC-0 (**C**) on $P_{Br}$ (black bars), $P_{NO3}$ (red bars) and $P_I$ (green bars). Nomenclature of α-hydroxy acid substitutions is explained in Methods. Data are Mean ± St.Dev. of $n > 4$ repeats from $N \geq 3$ independent oocyte batches, individual data points are shown as cyan circles. The statistical significance of the effects of the mutants on the permeability ratios of each ion (indicated by *) was evaluated with a one-sided Student's $t$-test with a Bonferroni correction (see "Methods"). Mean values and $p$ values are reported in Supplementary Table 1. Individual data points are shown grouped by ion in Supplementary Fig. 5. Raw data for **B, C** is included in the Source Data Files.

Fig. 4A), suggesting that the contribution of currents due to non-specific incorporation and endogenous channels is ≤ 10%. The α-hydroxy acids for glycine or glutamate are not commercially available, and α-hydroxy substitutions in hCLC-1 did not yield sufficient currents for reversal potential determination.

In the bCLC-k channel, $S_{ext}$ is lined by backbone amides of V166, M427 and F426, with F426 also lining $S_{cen}$ together with Y425 (Fig. 1B). α-hydroxy substitutions at $S_{ext}$, V166ω and M427μ, result in an altered selectivity order of $Br^- > Cl^- > NO_3^- > I^-$ while the others retain the WT order (Fig. 2B). The effects on $P_{Br}$ and $P_I$ are relatively small, with <50% changes relative to the WT values (Supplementary Fig. 6B) while effects on $P_{NO3}$ are large at all positions, highlighted by a ~250% $P_{NO3}$ increase in M427μ (Supplementary Fig. 6B). Thus, backbone amides contribute to the overall selectivity of bCLC-k, and amides lining $S_{ext}$ control the inter-anionic selectivity sequence.

We used the same approach to investigate the selectivity of the CLC-0 channel and found that mutating backbones lining $S_{cen}$ results in altered selectivity sequences, with F418φ not being able to discriminate between $NO_3^-$, $Br^-$ and $Cl^-$, and A417α showing an altered selectivity sequence of $Cl^- = NO3^- > Br^- > I^-$ (Fig. 2C), while the $S_{ext}$-lining V419ω substitution has a WT-like selectivity sequence (Fig. 2C). Thus, $S_{cen}$ appears to primarily determine selectivity in CLC-0, consistent with previous results[19,21,51]. Overall, the effects on $P_{Br}$, $P_{NO3}$ and $P_I$ are relatively small, with <50% changes relative to the WT channel (Supplementary Fig. 6A), likely reflecting the weaker inter-anionic selectivity of CLC-0 compared to bCLC-k (Fig. 1G). In both channels, backbone substitutions have parallel effects on the permeability ratios and on the conductivity of the various ions, estimated from the ratio of the currents at +80 mV in the foreign anion to that of $Cl^-$ (Supplementary Fig. 6C–H), indicating that interactions between backbone amides and the permeating ions determine binding and conduction.

### Pore-lining backbone amides play key roles in CLC-0 gating

Backbone mutations in the pore affect the G-V relationships of the single-pore gating process of CLC-0 in $Cl^-$ (Fig. 3A, Supplementary Fig. 7), with the A417α substitution inducing a left shift in the $G-V$ while

the F418φ and V419ω replacements cause a right-shift in $V_{1/2}$ (Fig. 3A, B). The direction of the $V_{1/2}$ shifts is preserved for all mutants in $Br^-$, $NO_3^-$ and $I^-$, although the magnitudes vary (Fig. 3B). Correlation between the effects on selectivity and those on gating is poor, as the A417α mutation strongly alters selectivity while having comparatively modest effects on gating and, conversely, the V419ω replacement has a WT-like selectivity profile and the largest effects on the $V_{1/2}$. Remarkably, these mutations have dramatic effects on the common-pore gating process (Fig. 3C), with A417α inverting its voltage dependence and V419ω resulting in a nearly constitutive phenotype (Fig. 3D). Thus, removal of a single hydrogen-bonding group in the CLC-0 pore affects the global rearrangements associated with slow gating[57], supporting the idea of a strong allosteric coupling between local and global rearrangements in this channel[46,47].

### Glu_ex modulates the role of backbone amides in selectivity

The differential roles of backbone amides in determining selectivity of CLC-0 and bCLC-k channels (Fig. 2) are surprising due to the overall structural conservation of CLC pores (Fig. 1). One obvious difference between these channels is that the highly conserved $Glu_{ex}$ of CLC-0 (E166) is replaced by an uncharged valine in bCLC-k (V166) (Fig. 1B, Supplementary Fig. 1E). Introducing $Glu_{ex}$ in the bCLC-k channel (V166E) or eliminating it from CLC-0 (E166A), has minor effects on selectivity with the V166E bCLC-K channel maintaining a WT-like sequence of $Cl^- > Br^- > NO_3^- - I^-$ (Fig. 4A), while the E166A CLC-0 mutant has a slightly altered sequence of $Cl^- - Br^- \geq NO_3^- > I^-$ (Fig. 4B).

We then introduced the backbone ester substitutions on the background of these mutants to test how $Glu_{ex}$ modulates the role of backbone amides in selectivity. Currents mediated by the V166E/Y425φ mutant were too small to obtain reliable results. The bCLC-k V166E/F426φ mutant, however, displays a marked preference for $Cl^-$ and does

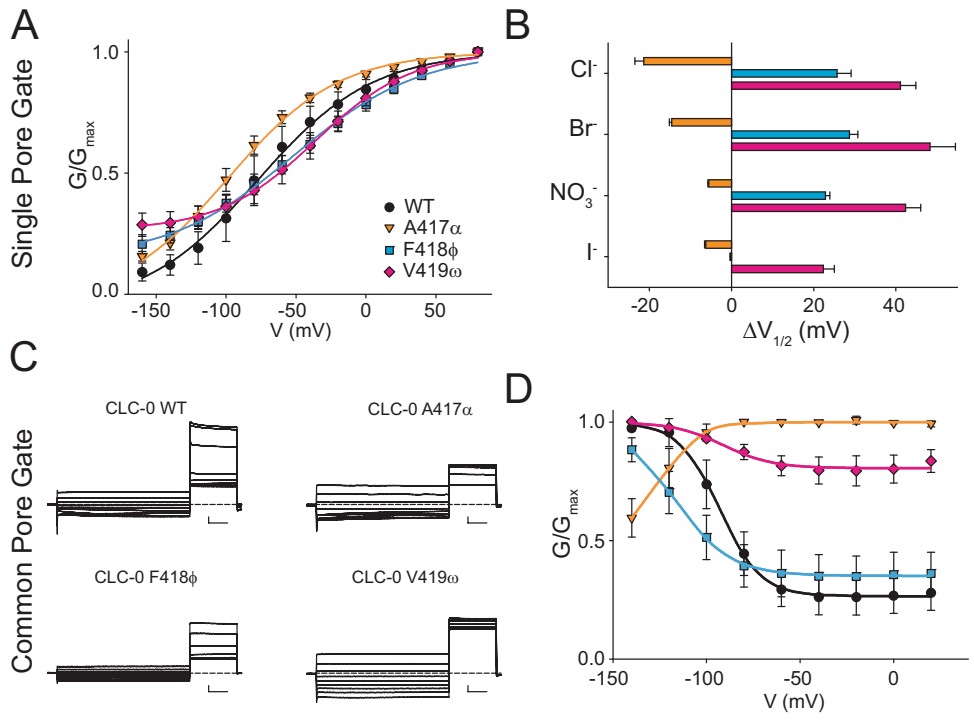

**Fig. 3 | Pore-lining backbone amides affect fast and slow gating in CLC-0.**
**A** Normalized $G-V$ curves for fast gate of CLC-0 WT (black), A417α (orange), F418φ (cyan) and V419ω (pink) in $Cl^-$. Solid lines are fits to Eq. 2. Values are mean ± St. Dev. of $n > 7$ repeats from $N \geq 3$ independent oocyte batches. **B** $\Delta V_{1/2} = (V_{1/2}^{mut} - V_{1/2}^{WT})$ of the normalized fast gate $G-V$ of WT and mutant CLC-0 in $Cl^-$, $Br^-$, $NO_3^-$ and $I^-$. Colors as in **A**. Errors represent the propagation of the uncertainty of the $V_{1/2}$ parameter

evaluated from the fits of the data in **A** and Supplementary Fig. 7. **C** Representative slow gate current traces of WT and mutant CLC-0. Scale bars indicate 0.5 μA and 1 s. **D** Normalized $G-V$ curves for slow gate of CLC-0 WT (black), A417α (red), F418φ (green) and V419ω (yellow) in $Cl^-$. Solid lines are fits to Eq. 2. Values are mean ± St. Dev. of $n > 7$ repeats from $N \geq 3$ independent oocyte batches. Raw data for **A**–**D** is included in the Source Data Files.

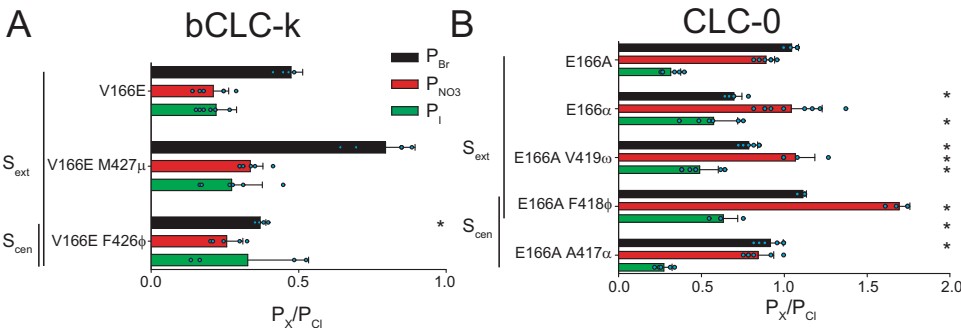

**Fig. 4 | A glutamate side chain at the Glu$_{ex}$ position modulates the role of S$_{cen}$ and S$_{ext}$ in anion selectivity. A, B** Effect of replacing backbone amides with ester oxygens lining the S$_{ext}$ and S$_{cen}$ sites in V166E bCLC-k (**A**) and E166A CLC-0 (**B**) on P$_{Br}$ (black bars), P$_{NO3}$ (red bars) and P$_I$ (green bars). Data are Mean ± St.Dev. of $n > 4$ repeats from $N \geq 3$ independent oocyte batches, individual data points are shown as cyan circles. The statistical significance of the effects of the mutants on the

permeability ratios of each ion (indicated by *) was evaluated with a one-sided Student's $t$-test with a Bonferroni correction (see "Methods"). Mean values and $p$ values are reported in Supplementary Table 1. Individual data points are shown grouped by ion in Supplementary Fig. 5. Raw data for **A, B** is included in the Source Data Files.

not discriminate among other anions resulting in an altered selectivity sequence of Cl$^-$>Br$^-$-I$^-$-NO$_3^-$ (Fig. 4A). In contrast, the V166E/M427μ mutant restores a WT-like selectivity sequence of Cl$^-$>Br$^-$>NO$_3^-$-I$^-$ (Fig. 4A), which was altered in the single M427μ mutant (Fig. 2B). In E166A CLC-0, the S$_{ext}$-lining E166α and E166A/V419ω substitutions have an altered selectivity sequence of NO$_3^-$-Cl$^-$>Br$^-$>I$^-$, suggesting an increased role for S$_{ext}$ in anion selectivity (Fig. 4B). Conversely, the S$_{cen}$ lining E166A/A417α mutant restores a WT-like selectivity sequence (Fig. 4B) that was lost in the single A417α mutant (Fig. 2C). Most strikingly, the E166A/F418ϕ mutation converts CLC-0 into a NO$_3^-$-selective channel with a selectivity sequence of NO$_3^-$>>Br$^-$>Cl$^-$>I$^-$ (Fig. 4B, Supplementary Fig. 6).

Overall, backbone manipulations in the absence of Glu$_{ex}$ have larger relative effects on ion selectivity than those in the presence of Glu$_{ex}$, in particular for P$_{NO3}$ (Supplementary Fig. 6A, B). In both channels, there is an imperfect correlation between the presence of a protonatable side chain at position 166 and the roles of amides lining S$_{cen}$ or S$_{ext}$ (Supplementary Fig. 6A, B). Together these observations suggest that the selectivity of CLC pores is modulated by the presence of a glutamate side chain that competes with permeant anions for occupancy of S$_{cen}$ and/or S$_{ext}$.

**Role of backbone amides in stabilizing anions in the bCLC-k pore**
To investigate the contribution of the protein residues, particularly the backbone amides, to stabilization of anions within the pore at a microscopic level, we simulated translocation of Cl$^-$, Br$^-$, I$^-$, or NO$_3^-$ through WT or M427μ bCLC-k channels[33] and calculated the potential of mean force (PMF) profiles associated with these single-ion permeation processes. For all anions, the PMF profiles show multiple local minima along the pore reporting on low-energy states of the ion where it establishes favorable interactions with the protein and local water molecules. Given the differences in sizes and H-bonding patterns of the anions (Fig. 5), small variations in the depth (~1–2 kcal/mole) and the exact positions (~1 Å) of these minima are observed.

The PMFs of WT channels show a global free energy minimum at S$_{ext}$ for all four inspected anions (marked with a green dashed line in Fig. 5A), highlighting this site as the most stable anion-binding region along the pore. At this site the anions are mainly coordinated by the backbone amides of K165, V166, and M427. Among other minima, which are all significantly shallower than S$_{ext}$, noticeable are the one at or around S$_{cen}$ (marked with a purple dashed line in Fig. 5A) in which the anions are coordinated by the backbone amide of Y425, and one in between S$_{ext}$ and S$_{cen}$ (marked with an orange dashed line in Fig. 5A) where the anions are coordinated by the backbone amides of G167 and Y425. These results indicate that the backbone amide interactions play

a key role in stabilizing the anions in all three minima (Fig. 5C). At all three sites, direct interactions between the anions and water molecules contributes to their stabilization. The M427μ replacement disrupts anion binding to S$_{ext}$ (Fig. 5B) resulting in significant reduction of the well depth at this site or its complete disappearance as ions lose their favorable coordination with the amide hydrogen. Beyond local effects on S$_{ext}$, this mutation also affects the pattern of hydration, not only inside the constricted region, but also in the region between S$_{ext}$ and S$_{cen}$, by altering the orientation of the water molecule coordinating to M427 in the WT protein (Fig. 5D). Furthermore, coordination analysis on the permeant anions suggests that the M427μ mutation affects the hydration and ion-protein interaction patterns beyond the S$_{ext}$ site (Supplementary Fig. 8), which we believe could account for the PMF changes observed in other regions of the pore after the mutation. Previous MD studies showed that hydration and protein coordination of ions along the pore can affect the energetics of the ion permeation process in other channels[58,59]. These results exemplify the critical contribution of backbone amide coordination for the anions within the pore. While this treatment does not represent the prevailing multi-ion permeation mechanism in these channels, it probes the environment of the anions in the pore and provides a reliable approximation for the energetics experienced by them.

## Discussion
Ion channels employ diverse molecular strategies to enable the rapid and selective passage of ions across biological membranes. While cation selectivity is relatively well understood, the molecular bases of anion selectivity remain poorly characterized. Indeed, many anion channels are more permeable to non-physiological ions, such as I$^-$ or SCN$^-$, than to the physiologically abundant Cl$^-$ [9–12,14] and most small-molecule compounds designed to bind and transport anions share a similarly poor selectivity profile[60–62], highlighting our limited understanding of the fundamental mechanisms of anion selectivity. Unique among anion channels, the CLCs specifically select for Cl$^-$ over other anions. Past work suggested that the CLC preference for Cl$^-$ is determined by the specific interactions with a pore-lining serine side chain on the C-D loop, as substitutions at this position confer selectivity for other anions[19–21,23,50,52]. However, this mechanism was recently questioned as this loop adopts a different conformation in kidney-type CLC-k channels[33,53] and its functional role is not conserved in the muscle-type CLC-1 channel (Fig. 1G). Thus, the determinants of the conserved Cl$^-$ selectivity of the CLCs remain unknown.

Ions in the CLC permeation pathway form hydrogen bonds with backbone amides lining S$_{cen}$ and S$_{ext}$[33,39] whose position is well-conserved among CLC proteins (Fig. 1, Supplementary Fig. 1). We used

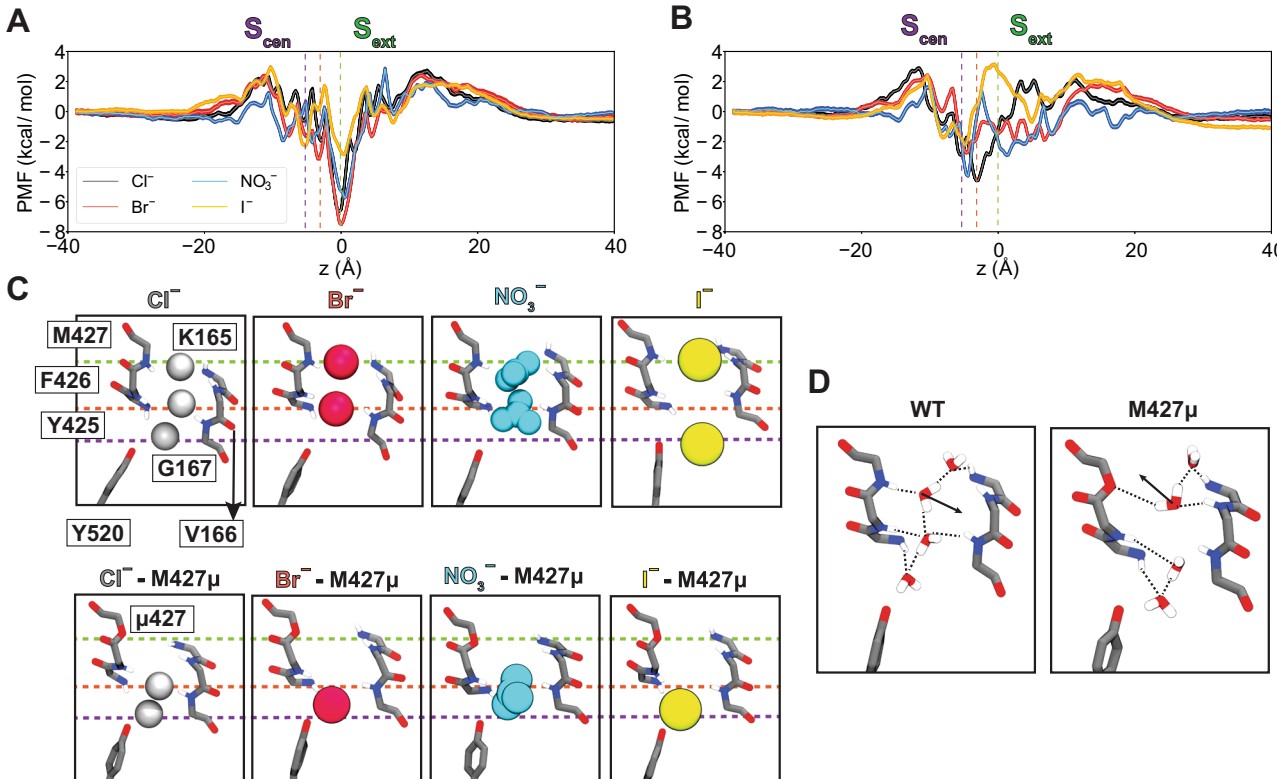

**Fig. 5 | Anion-backbone interaction along the permeation pathway in WT and M427μ bCLC-k. A** Single-occupancy PMF calculated along the ion permeation pathway for $Cl^-$, $Br^-$, $I^-$, and $NO_3^-$ in WT bCLC-k. The $z$ positions for the ions are specified relative to $S_{ext}$ ($z = 0$). The positions of $S_{ext}$, an intermediate binding region, and $S_{cen}$ are indicated by the green, orange, and purple vertical dashed lines, respectively. PMFs are aligned by their energy in the bulk solution. B) Single-occupancy PMF calculated along the ion permeation pathway for $Cl^-$, $Br^-$, $I^-$, and $NO_3^-$ in the M427μ mutant of bCLC-k. The uncertainty error in (**A**) and (**B**) is calculated based on Monte Carlo bootstrapping and shown in shaded colors (almost unnoticeable). **C** The selectivity filter of bCLC-k along with the positions of all major binding sites (based on the PMFs, panels (**A**) and (**B**)) for different anions depicted in vdW for both WT and the M427μ mutant ($Cl^-$ in white, $Br^-$ in red, $NO_3^-$ in blue, and $I^-$ in yellow). The three dashed lines correspond to those drawn in panels (**A**) and (**B**). **D** Comparison of hydration patterns in the selectivity filter of WT and M427μ bCLC-k, highlighting the shift in the orientations of water molecules.

atomic mutagenesis to site-specifically replace these putative hydrogen-bond donor, backbone amides with an ester oxygen that cannot engage in hydrogen bonds with the permeating anions[55,56]. We found that targeted removal of individual pore-lining amides substantially degrades inter-anionic discrimination in both CLC-0 and bCLC-k, resulting in channels with weakened $Cl^-$ preference (Figs. 2 and 4). Indeed, elimination of a single hydrogen bond within the selectivity filter can increase $P_{NO3}$ up to 250% and $P_I$ up to 200% (Supplementary Fig. 6), while effects on $P_{Br}$ are generally more modest, consistent with the idea that $Br^-$ is a faithful substitute for $Cl^-$[31,38,43,48]. It is possible that the non-specific incorporation of amino acids and/or endogenous channels dampen the effects of backbone mutations on selectivity. However, currents recorded in oocytes injected with unconjugated tRNA are ≤ 10% of those from tRNA conjugated to the α-hydroxy acids (Supplementary Fig. 4A), suggesting contributions of endogenous currents are relatively small and will not affect the observed trends. These errors could be higher for $I^-$ currents, as this ion blocks CLCs but is more permeable through other $Cl^-$ channels. Overall, the magnitude of the observed effects on selectivity is relatively small, likely reflecting the weaker interanion selectivity of $Cl^-$ channels. Nonetheless, it is remarkable that single substitutions of single backbone atoms can alter the anion selectivity sequences of both bCLC-k and CLC-0 (Figs. 2 and 4). Notably, equivalent manipulations do not affect the selectivity sequence of $K^+$ channels[63,64], supporting the idea that interactions between backbone amides and permeating anions are specifically important for selectivity in CLCs.

The specific roles of backbone amides depend on the specific channel environment, as equivalent substitutions do have the same effects in CLC-0 and bCLC-k (Fig. 2). Further, not all backbone amides play comparable roles in selectivity, for example V166ω and M427μ render bCLCl-k less selective whereas F426φ increases its preference for $Cl^-$ (Fig. 2). These differences could arise, at least in part, from contributions of variable pore-lining side chains. Indeed, the roles of backbone amides in anion selectivity is enhanced in channel constructs lacking a $Glu_{ex}$ side chain (E166A CLC-0 and WT bCLC-k) relative to those with a glutamate at this position (WT CLC-0 and V166E bCLC-k) (Figs. 2, 4, Supplementary Fig. 6), and side chains also play roles in selectivity in some CLCs (Fig. 1)[19–21,23,51,53]. Thus, we propose that $Cl^-$ specificity in CLC channels and transporters arises from the combined contributions of pore-lining backbone amides and side chains. Together, our data suggests that the structurally conserved orientation of backbone amides provides optimized coordination for the permeating $Cl^-$ ions while pore-lining side chains, that are variable in sequence and orientation, modulate the contributions of individual backbone amides as well as directly coordinating ions in the pore, as suggested by the poorly conserved role of the $Ser_{cen}$ side chain in CLC selectivity. Further work is required to dissect the contributions of individual structural elements to anion selectivity.

It is important to consider that ion permeation through the CLCs is a multi-ion process[46], which raises the possibility that ion selectivity could also be determined by multi-ion occupancy of the pore. However, several lines of evidence suggest this is not the case. First, the CLC channels and exchangers share the same selectivity sequence despite

having different mechanisms of ion transport. Second, in CLC-ec1 – a CLC transporter that shares the same ion selectivity as the CLC-0 and bCLC-k channels studied here – the selectivity of ion binding and transport coincide[18,19] and ion binding to the different sites is largely unaffected in single or multi-ion configurations[19,38]. Together, these observations suggest that the selectivity properties of the CLC Cl⁻ permeation pathway are determined by the interactions of single ions with the pore and thus can be evaluated using single-ion PMF calculations (Fig. 5). Our free energy calculations highlight the involvement of backbone-ion coordination in the stabilization of the ions in the channel's selectivity filter and at the $S_{ext}$ site, where we find that mutating the pore-lining backbone amide of M427μ directly and significantly destabilizes ion binding. Notably, our simulations show this manipulation also results in longer-range effects as it perturbs the hydration pattern of the pore (Fig. 5D). Finally, we found that substituting pore-lining backbone amides affects CLC-0 gating, with particularly marked effects on common gate activation. This supports the idea of tight allosteric coupling between ions permeating through the pore and the local and the global rearrangements that respectively underlie single- and common-pore gating processes in this channel[46,47].

Our MD simulations suggest that in CLC-k the ions are most strongly bound at the $S_{ext}$ site. In the backbone mutant M427μ, $S_{ext}$ becomes unstable and the binding positions of ions and water near $S_{cen}$ are shifted down with a clear change in their orientations, factors that likely result in the channel's loss of selectivity for Cl⁻. The proposed mechanism, that ion selectivity is primarily determined via interactions with backbone elements, is reminiscent of the mechanism for selectivity in K⁺ channels[1,2]. The channel's structure is optimized to provide an ideal coordination shell to the permeating ion via interaction with its backbone, where the choice of carbonyls or amides facilitates ions of different charge.

## Methods

### In vitro cRNA transcription

RNAs for all CLC-0 and bCLC-k wild-type and mutant constructs were transcribed from a pTLN vector using the mMessage mMachine SP6 Kit (Thermo Fisher Scientific, Grand Island, NY)[46,52,65]. For final purification of cRNA the RNeasy Mini Kit (Quiagen, Hilden, Germany) was employed. RNA concentrations were determined by absorbance measurements at 260 nm and quality was confirmed on a 1% agarose gel.

### tRNA misacylation

For nonsense suppression of CLC-0 and bCLC-k TAG mutants in *Xenopus laevis* oocytes, THG73 and PylT tRNAs have been employed. THG73 was transcribed, folded and misacylated as previously described[66]. PylT was synthetized by Integrated DNA Technologies, Inc. (Coralville, IA, USA), folded and misacylated as previously described[67]. Ala-, Met-, Phe-, Val-, α-hydroxy Ala- (α), α-hydroxy Met- (μ), α-hydroxy Phe- (φ) and α-hydroxy Val-pdCpA (ω) substrates were synthesized according to published procedures[67].

### Nonsense suppression to replace amino acids with α-hydroxy acid

The nonsense suppression method was used to site-specifically replace amino acids with pore-lining backbone amides with their α-hydroxy acid equivalents[55]. This atomic manipulation substitutes the backbone NH group with an oxygen atom, eliminating the ability of the backbone to function as H-bond donor without altering side chain properties (Fig. 2A), converting the peptide bond into an ester bond. These bonds have similar lengths, angles, preference for a trans geometry, and comparably high energy barrier for rotation[55,56]. Incorporation of the α-hydroxy acids at the positions tested in bCLC-k (V166, Y425, F426, M427) and CLC-0 (E166, A417, F418, V419) resulted in currents that were at least 9-fold higher than those recorded in oocytes injected with non-

acetylated control tRNA (Supplementary Fig. 4A). We indicate mutations to α-hydroxy acids with their Greek letter counterpart: α for α-hydroxy alanine, ω for α-hydroxy valine, φ for α-hydroxy phenylalanine and μ α-hydroxy methionine. Incorporation of WT amino acids resulted in channels with WT-like properties (Supplementary Fig. 4B, C). Finally, insertion of φ at position F161 (F161φ) in CLC-0, a pore-lining residue located near $Glu_{ex}$ (E166) but not involved in ion binding, resulted in WT-like selectivity (Supplementary Fig. 4D, E). These results indicate that effects on selectivity specifically reflect the incorporation of α-hydroxy acids at the targeted positions.

We were not able to test the role of the following residues: (i) G164 (bCLC-k and CLC-0) because the α-hydroxy glycine acylated to the suppressor tRNA was very prone to hydrolysis impeding the incorporation; (ii) K165 (bCLC-k), R165 (CLC-0), and E166 as α-hydroxy acids could not be synthetized, and (iii) we used α-hydroxy phenylalanine (φ) at position Y425 (bCLC-k) as the incorporation of α-hydroxy tyrosine was not successful. Phenylalanine was used as control in this case (Supplementary Fig. 4B). Currents associated with the V166E Y425φ mutant were too small to be analyzable. The nonsense suppression approach did not result in analyzable currents of CLC-1, CLC-5 or CLC-7.

### Protein expression in *Xenopus laevis* oocytes and two electrode voltage clamp (TEVC) recordings

*Xenopus laevis* oocytes were purchased from Ecocyte Bio Science (Austin, TX, USA) and Xenoocyte (Dexter, Michigan, USA) or kindly provided by Dr. Pablo Artigas (Texas Tech University, USA, protocol # 11024). For conventional CLC expression, the following injection and expression conditions have been used: for CLC-0, 0.1–5 ng cRNA were injected and currents were measured ~6–24 h after injection; for CLC-1, ~2 ng cRNA were injected and currents were measured ~ 24 h after injection; ~0.1 ng of each, CLC-K and Barttin cRNA, were coinjected and currents were measured the day after injection. For nonsense suppression of CLC-0 and bCLC-k constructs, cRNA and misacylated tRNA were coinjected (up to 25 ng of cRNA and up to 250 ng of tRNA per oocyte) and currents were recorded 6–24 h after injection.

TEVC was performed as described[19,68]. In brief, voltage-clamped chloride currents were recorded in ND96 solution (in mM: 96 NaCl, 2 KCl, 1.8 CaCl2, 1 MgCl2, 5 HEPES, pH 7.5) using an OC-725C voltage clamp amplifier (Warner Instruments, Hamden, CT). Ion substitution experiments were performed by replacing the 96 mM NaCl in the external solution with equimolar amounts of NaBr, NaNO₃ or NaI. Data was acquired with Patchmaster (HEKA Elektronik, Lambrecht, Germany) at 5 kHz and filtered with Frequency Devices 8-pole Bessel filter at a corner frequency of 2 kHz. Analysis was performed using Ana (M. Pusch, Istituto di Biofisica, Genova), Sigmaplot (SPSS Inc.) and Prism (GraphPad, San Diego, CA, USA). For each substitution we recorded currents from oocytes injected with unconjugated tRNA (Fig. 3 Supplementary 1A). This current, I(tRNA), reflects a combination of the contributions of CLC channels with non-specific incorporation of conventional amino acids and of the endogenous currents. In all cases, the ratio of the currents measured in oocytes injected with tRNA conjugated to the UAA, I(UAA), to I(tRNA) is >9 (Fig. 3 Supplementary 1A). This suggests that the contribution of currents due to non-specific incorporation and endogenous channels is ≤10% and thus will not affect the trends of the observed effects. It is possible that the contribution to error could be higher for I⁻ currents, as this ion blocks CLCs but is more permeable through other Cl⁻ channels.

Oocytes were held at a resting potential of −30 mV. For CLC-0 two different recording protocols have been used to distinguish single-pore from common-pore gating. During the single-pore gating protocol the voltage was stepped to +80 mV for 50 ms and then a variable voltage from −160 mV to +80 mV increasing in 20 mV steps was applied for 200 ms, followed by a 50 ms pulse at −120 mV for tail current analysis. For CLC-0 common-pore gating, 7 s voltage steps

from +20 mV to −140 mV have been applied in −20 mV increments followed by a 2.5 s + 60 mV post pulse for tail current analysis. For bCLC-k the voltage was stepped to −30 mV for 20 ms and then a variable voltage from −80 mV to +80 mV increasing in 10 mV steps was applied for 150 ms, followed by a 20 ms pulse at −30 mV. For CLC-1 the voltage was stepped to +80 mV for 100 ms and then a variable voltage from −160 mV to +80 mV increasing in 20 mV steps was applied for 200 ms, followed by a 100 ms pulse at −100 mV for tail current analysis.

## Analysis of electrophysiological recordings

Permeability ratios were determined by measuring the change in reversal potential, $\Delta V_{rev}$, recorded upon substituting the external anion and using the Goldman-Hodgkin-Katz equation[15] as

$$\Delta V_{rev} = \left( V_{rev}^2 - V_{rev}^1 \right) = \left( \frac{RT}{zF} \ln \frac{P_{Cl}[Cl]_{ex}^2 + P_X[X]_{ex}^2}{P_{Cl}[Cl]_{in}} \right) \\ - \left( \frac{RT}{zF} \ln \frac{P_{Cl}[Cl]_{ex}^1}{P_{Cl}[Cl]_{in}} \right) = \frac{RT}{zF} \ln \frac{P_{Cl}[Cl]_{ex}^2 + P_X[Cl]_{ex}^1}{P_{Cl}[Cl]_{ex}^1} \quad (1)$$

Where $R$, $T$, $F$ and $z$ have the usual meaning. The assumption that $[Cl]_{in}$ did not change during successive perfusions was validated by bracketing recordings in $Br^-$ and $NO_3^-$ with a return measurement in external $Cl^-$ and ensuring that $V_{rev}$ did not shift by more than 3 mV. Thus, the sequence of experiments was $Cl^-(1)$, $Br^-$, $Cl^-(2)$, $NO_3^-$, $Cl^-$ (3), $I^-$ (Supplementary Fig. 3). In some cases, the order of $Br^-$ and $NO_3^-$ was inverted, but no differences were detected. $I^-$ was kept as the last ion tested due to its slow washout from oocytes. To simplify notation, throughout the text we indicate the relative permeability ratios of $Br^-$, $NO_3^-$, and $I^-$ as $P_{Br}$, $P_{NO3}$ and $P_I$ with the understanding that these values represent the relative permeability ratios of these anions to that of $Cl^-$, $P_{Br, NO3, I}/P_{Cl}$.

To estimate the voltage dependence of WT and mutant CLC-0, tail current analysis was performed, and data was fit to a Boltzmann function of the form:

$$P_o = P_{min} + \frac{(1 - P_{min})}{1 + e^{[(V_{0.5} - V)/k]}} \quad (2)$$

where $P_o$ is the open probability as a function of voltage and is assumed to reach a value of unity at full activation. $P_{min}$ is the residual open probability independent of voltage. $V_{0.5}$ is the voltage at which 50% activation occurs, and $k = RT/zF$ is the slope factor, $R$ is the universal gas constant, $T$ is temperature in $K$, $F$ is the Faraday constant, and $z$ is the gating charge.

## Statistical analysis

All values are presented as mean ± S.Dev. as indicated in the pertinent figure legends. One-sided Student's $t$-test was performed to determine statistical significance of effects. We compared the effects of each α-hydroxy substitution to its parent construct. Thus, each single mutant was compared to the WT channel constructs whereas each double mutant was compared to the corresponding E166A or V166E mutants. The threshold for significance of the one-sided Student's t-test was assigned using a threshold of $p = 0.05$ and a Bonferroni correction. Thus, the significance thresholds are: CLC-0 WT $p < 0.007$, CLC-0 E166A $p < 0.006$; bCLC-k WT $p < 0.0055$; bCLC-k V166E $p < 0.01$ and CLC-1 $p < 0.05$.

## Statistics and reproducibility

Functional experiments were repeated 4+ times from 3+ independent oocyte batches.

## Simulation systems setup

For the bovine bCLC-k channel, the cryo-EM structures[33] (pdb:5TQQ) were used as the structural model for all the MD simulations and free energy calculations. Two unstructured loop regions missing in the cryo-EM structures (residue 258-276 and 454-456) were modeled using SuperLooper[69]. The resulting models were embedded in lipid bilayers consisting of 80% POPC and 20% cholesterol and solvated with 0.15 M of NaCl and TIP3P water[70] using CHARMM-GUI MEMBRANE BUILDER[71]. The dimension for the simulated systems was $150 \times 150 \times 130$ Å$^3$.

The simulation system was energy-minimized for 10,000 steps, followed by two steps of 1-ns relaxation. The simulation system was then subjected to 1 ns of NPT initial equilibration with the standard protocol described in the CHARMM-GUI MEMBRANE BUILDER, which involves gradually releasing positional and dihedral restraints on the protein and lipid molecules. Thereafter, 10 ns of NPT equilibration with dihedral restraints ($k = 100$ kcal/mol/rad$^2$) on the protein secondary structure were performed. After equilibration, the simulation system was equilibrated for another 250 ns without any restrains. This equilibrated system was used for all the free energy calculations.

## Simulation protocols

All simulations were carried out with NAMD 2.13[72,73], using CHARMM36m protein[74] and CHARMM36 lipid[75] parameters. The SHAKE algorithm[76] was employed to constrain bonds involving hydrogens to allow 2-fs timesteps for the integrator. A constant temperature of 310 K was maintained by Langevin thermostat[77] with a damping coefficient of 1 ps$^{-1}$. Nosé-Hoover Langevin piston[78] with a period of 200 ps and a decay time of 50 ps was employed to maintain constant pressure at 1 atm. Periodic boundary conditions and a nonbonded cutoff of 12 Å (with a 10 Å switching distance and using vdW force switching) were used. Long-range electrostatics were calculated using the particle mesh Ewald method[79] with 1-Å grid spacing. Bonded interactions and short-range nonbonded interactions were calculated every timestep (2 fs). The pairs of atoms whose interactions were evaluated (neighborhood list) were updated every 20 fs. A cutoff (13.5 Å) slightly longer than the nonbonded cutoff was applied to search for interacting atom pairs. Publicly available software package VMD was used for the analysis and visualization of the molecular system (https://www.ks.uiuc.edu/Research/vmd/).

## Free energy calculations

The free energy profiles, or the potential of mean force (PMF), of ion translocation through the permeation pore of bCLC-k were calculated using an enhanced sampling technique, umbrella sampling (US)[80]. In the US simulations, the reaction coordinate was chosen to be the $z$ position (along the membrane normal) of the restrained ion relative to $S_{ext}$ (set to $z = 0$), as the permeation pathway near the selectivity filter is roughly parallel to the membrane normal (aligned with the $z$-axis). To restrain the ion movement through the selectivity filter, the $xy$ coordinates of the ion were confined by a cylindrical half-harmonic wall ($k = 10$ kcal/mol/Å$^2$) with a radius of 30 Å centering around the axis of the permeation pathway. For each ion the conduction pathway was divided into 80 umbrella windows with 1 Å interval and ranging from $z = -40$ Å to $z = 40$ Å, assuring the ion was in solution at each end. In each window, the ion was harmonically restrained along the reaction coordinate ($k = 5$ kcal/mol/Å$^2$), and initially equilibrated for 1 ns. Production sampling over each window was then done for 10 ns. The obtained distributions were then unbiased and combined using the weighted histogram analysis method (WHAM)[81] to obtain the PMF of the ion movement along the pore axis. The convergence of each PMF was examined by constructing the PMF after 5 to 10 ns of sampling with 1 ns interval. All the obtained PMFs remain unchanged after 9 ns of sampling, indicating a good degree of convergence (Fig. 5 Sup. 2). The final PMFs are constructed with 10 ns of sampling and uncertainty errors are calculated based on Monte Carlo bootstrapping.

**Reporting summary**

Further information on research design is available in the Nature Portfolio Reporting Summary linked to this article.

## Data availability

All data, constructs and electrophysiological traces are available on request. MD simulation files are available at: https://doi.org/10.5281/zenodo.7317338. Source data are provided with this paper.

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

## Acknowledgements

The authors thank members of the Accardi lab for helpful discussions. The work was supported by National Institutes of Health (NIH) grants R01-GM128420 (to A.A.), R01-GM106569 and NINDS R24 NS104617 (to C.A.A.), R01-GM123455 and P41-GM104601 (to E.T.). Simulations in this study have been performed using allocations at National Science Foundation Supercomputing Centers (XSEDE grant number MCA06N060), and the Blue Waters Petascale Computing Facility of National Center for Supercomputing Applications (NCSA) at University of Illinois at Urbana-Champaign, which is supported by the National Science Foundation (awards OCI-0725070 and ACI-1238993) and the State of Illinois.

## Author contributions

L.L., K.L., S.D., C.A.A., E.T. and A.A. designed experiments; L.L., K.L., S.D., E.F. and J.D.G. performed experiments; L.L., K.L., S.D., E.T. and AA analyzed the data; A.A. prepared an initial draft and all authors edited the manuscript.

## Competing interests

The authors declare no competing interests.
