## [Peer Review File · Nature Communications]

Backbone amides are determinants of Cl⁻ selectivity in CLC ion channelsReviewers' Comments:

Reviewer #1:

Remarks to the Author:

Ion transport in CIC chloride channels and exchangers occurs at a structurally conserved selectivity filter consisting of three chloride binding sites. Using backbone mutagenesis and molecular dynamics simulations, the authors propose that anion selectivity in CIC channels is in part contributed by mainchain amides at the external and central anion binding sites, which might stabilize the bound anions via hydrogen bonding. While the present data provide some support for this mechanism, the following comments and inconsistencies will need to be carefully addressed.

1. While there are changes in the backbone mutants' anion selectivity in some instances, the changes are generally small (<1.5-fold compared to WT) with the exception of Pno3 in the bCIC-k backbone mutants and the spread of the data are in some instances large. These moderate changes might be further skewed by the presumably lower expression in some mutants as hinted in Fig. 2 S1 especially for the least permeant ions. For instance, in CIC-0, the most notable shift is the Pno3 of F418φ, which is still less than 1.5-fold compared to WT. Although there can be compensatory effects/redundancy amongst the amides, I would be cautious in not overstating the impact of these amide hydrogens on anion selectivity. Please tone down the statements in the title, abstract, results, and discussion.

2. The authors should address the modest effect of S189P in hCIC-1 given the proposed role of hydrogen bonding in anion selectivity and that this residue is a hydrogen bond donor that is depicted in Fig. 1 to be capable of interacting with the bound anions.

3. The authors should provide rationales and/or interpretations of the directionality of the shifts in the quantified properties. While changes in the mutant properties indicate the involvement of the residues/backbone, rationales/interpretations of how these changes are brought about are required to support the proposed mechanisms, which are not present in the manuscript.

4. The dependence of the amide effects on the gating glutamate is not supported by the data in the current manuscript. For instance, the effects of backbone amide substitutions are qualitatively similar on the WT and V166E backgrounds in bCIC-k and on the WT and E166A backgrounds in CIC-0, with the exception of Pno3 in M427μ for bCIC-k and Pbr of F418φ in CIC-0. These results are therefore not consistent with a dependence on the gating glutamate, which might in fact be expected if the underlying mechanisms involve primarily partial charges as suggested by the amide-to-ester mutations. Please provide additional data and/or analysis to support this claim and the resulting interpretation in the discussion or modify the relevant texts accordingly.

5. The PMF profiles for Cl, Br, and NO₃ in bCIC-k overlap substantially across the selectivity filter, which were calculated for single-ion occupancy. The authors should explain how this is consistent with the >3-fold selectivity of Cl versus NO₃ and iodide and address the limitations of either the experimental or computational approaches and the potential role of multi-ion occupancy in anion selectivity. Please also include the data for iodide as it has the lowest relative permeability amongst the anions tested.

6. From the simulations, no clear evidence is presented on how disrupting the M427 amide can lead to an increase in the relative permeation of Br and NO₃ if amide-stabilized ion occupancy is a critical determinant for anion selectivity. The energy minima for Br are the most severely perturbed yet Pbr is increased in the experiments shown in Fig. 2B and the energy minima for NO₃ are similar to that of Cl yet Pno3 is 2-fold less than that of Cl in Fig. 2B. The authors should also include the perturbation of ion occupancy for Br and NO₃ in Fig. 5C as Br binding is expected to be more substantially weakened by the mutation than Cl according to the PMF profiles shown in Fig. 5A and B.

7. Given the spread of the data and that the changes are modest in most cases, the authors should

explicitly include statistical comparison in Fig. 1-4. Please include the individual data points to help readers visualize the underlying spread of the data in Fig. 1-4.

Reviewer #2:

Remarks to the Author:

This interesting work dissects the anion selectivity of CLC channels, resolving the seeming contradiction between the role of Scen in anion discrimination in various CICs across taxa and the structural and mutagenesis results obtained from hCIC-Ka. The authors suggest that the backbone amides of the pore-lining amino acid residues act as H donors and this hypothesis is tested by incorporation of alpha-hydroxy acids by nonsense suppression; the resulting ester bonds have similar dimensions as C-N bonds, but are unable to serve as H donors. These changes in backbone of Scen and Sext residues cause changes in selectivity sequence for both CIC-0 and CIC-Ka, clarifying the nature of inter-anion discrimination of CLCs. This nice piece of work significantly advances our understanding of CIC channel permeation and selectivity.

Experiments are well-designed, the paper is very well written, and I have only minor criticism concerning data presentation:

1. If the same traces are shown in e.g. Fig 1E and Fig 1-Suppl. 2, it should be stated in the legend.
2. It is better to show averaged traces and averaged I-V relationships rather than rather than obtained from a single cell (even though I do not doubt that they are indeed representative).
3. It would be nice to see all data points with bars for averages and error bars depicting SD rather than SEM for PX/PCI. This way the variability of measurements is not masked by the sample size.
4. It may be nice to have WT values for reference in Fig2B,C and Fig4.

Other minor points:

1. Fig 2 legend: should be PNO3 rather than PNO
2. P.15: The nonsense suppression method to site-specifically replace amino acids with pore-lining backbone amides with their α -hydroxy acid equivalents. Sentence has no verb.

Reviewer #3:

Remarks to the Author:

This manuscript examines the origins of Cl⁻ selectivity over other anions in the family of CLC channels using a combination of unnatural amino acid mutagenesis, electrophysiology and molecular dynamics simulations. For most biological channels, anion selectivity is not critical, but CLC channels are odd in that most favour Cl⁻ permeation over other anions in contrast to the Hofmeister selectivity series often seen for other proteins. The topic is therefore of considerable interest, not only for understanding this class of protein, but also for appreciating the general strategies that can be used to develop selective ion transport. Overall, this is a well constructed study, and the use of unnatural amino acids to probe the role of backbone amides is well conceived and interesting.

A defining structural difference between the bCLC-k and h-CLC-1 channels is the position of the central Serine side chain (Ser_{cen}), and this is discussed as a source for functional differences. However, it is possible that in physiological conditions this side chain could be dynamic and move between these positions and that the structures may only be capturing one state in their conditions (eg due to low temperature). Because this is the basis for much of the mechanistic differences proposed for the channels, I would like to see a discussion as to whether this is a robust difference between the two channels. Is there any evidence of side chain motion in the MD simulations of bCLC-k conducted here?

For the bar graphs of selectivity ratio in Fig 2 (B,C) and Fig 4 a statistical analysis to test whether the changes between the WT and mutants are statistically significant is needed. Because this is the key

experimental data, it is important to be clear as to what changes are significant.

The MD simulations suggest that the external binding site, Sext, has the strongest binding affinity. Can the changes in selectivity seen between the different channels *especially for Cl⁻ vs Br⁻) be explained by the presence or absence of this site? For example, if this site had the greatest selectivity, then removing it (eg M427u) could remove the Cl⁻ to Br⁻ selection. To better explore this, I would really like the selectivity of this site to be examined using the MD simulations with FEP or TI calculations. A clear result from the MD simulations as to the relative selectivity of different locations along the pore would make the origins of selection more clear.

I recommend plotting the ion hydration and protein coordination number as a function of position along the pore. This can be determined from the MD simulations used to make the PMFs and can help to understand the origin of the energy wells and the differences between the ions. The degree of ion dehydration is a well discussed mechanism for generating selection between cations, and has been shown to similarly play a role for anions (Richards et al Small, 8: 1701-1709, 2012; Phys Chem Chem Phys, 14: 11633-11638, 2012). This information could assist in clarifying the origins of selection in the current study.

I recommend being a bit more nuanced in comparing the role of the backbone amides, and the serine side chain in defining selectivity. It may be the case that both play some role, rather than being just one or the other.

Fig 2. In the legend, please explain the meaning of the dashed lines in panels B and C (I did work it out, but better to be explicit)

Fig 4. Include the WT values for reference

Reviewer #1 (Remarks to the Author):

Ion transport in ClC chloride channels and exchangers occurs at a structurally conserved selectivity filter consisting of three chloride binding sites. Using backbone mutagenesis and molecular dynamics simulations, the authors propose that anion selectivity in ClC channels is in part contributed by mainchain amides at the external and central anion binding sites, which might stabilize the bound anions via hydrogen bonding. While the present data provide some support for this mechanism, the following comments and inconsistencies will need to be carefully addressed.

1. While there are changes in the backbone mutants' anion selectivity in some instances, the changes are generally small (<1.5-fold compared to WT) with the exception of P_{NO_3} in the bClC-k backbone mutants and the spread of the data are in some instances large. These moderate changes might be further skewed by the presumably lower expression in some mutants as hinted in Fig. 2 S1 especially for the least permeant ions. For instance, in ClC-0, the most notable shift is the P_{NO_3} of F418 ϕ , which is still less than 1.5-fold compared to WT. Although there can be compensatory effects/redundancy amongst the amides, I would be cautious in not overstating the impact of these amide hydrogens on anion selectivity. Please tone down the statements in the title, abstract, results, and discussion.

We thank the reviewer for bringing up this important point. We agree that the absolute values of the observed effects on selectivity are not very large, likely reflecting the poor anion selectivity of Cl^- channels compared to K^+ or Na^+ channels. For this reason, in our analysis we focused mainly on mutants that alter the order of the selectivity sequence, rather than emphasizing the absolute values of these changes. Overall, we find it remarkable that replacement of a single atomic bond alters the rank order of selectivity. Notably, equivalent manipulations do not affect the selectivity sequence of K^+ channels (Lu et al., 2001; Valiyaveetil et al., 2006), supporting the idea that interactions between backbone amides and permeating anions are important for selectivity in CLCs. Further, we note that not all backbone amides play equivalent roles and that their roles are modulated by the pore environment, such as by presence of a negatively charged side chain at position 166 of CLC-0 and bCLC-k. We have extensively revised Discussion (Pg. 13-14) to explicitly discuss these important points.

2. The authors should address the modest effect of S189P in hClC-1 given the proposed role of hydrogen bonding in anion selectivity and that this residue is a hydrogen bond donor that is depicted in Fig. 1 to be capable of interacting with the bound anions.

We agree with the reviewer that the lack of effect of S189P CLC-1 is surprising. We are not sure of the basis for this. It is possible that the S_{cen} site in this channel might be weaker than in other family members, as suggested by the lack of density at this site in the cryoEM structure (Fig. 1C) (Park and MacKinnon, 2018; Wang et al., 2019), so that it plays a reduced role in selectivity. Alternatively, it is possible that the C-D loop in this channel is more flexible than the structures suggest (as suggested by Reviewer 3), so that S189 points towards S_{cen} only for a fraction of the time, thus reducing the importance of this hydrogen bond to ion binding. We now discuss this in the Results section on page 6.

3. The authors should provide rationales and/or interpretations of the directionality of the shifts in the quantified properties. While changes in the mutant properties indicate the involvement of the residues/backbone, rationales/interpretations of how these changes are brought about are required to support the proposed mechanisms, which are not present in the manuscript.

We thank the reviewer for this suggestion. However, our data suggest the effects at each site depend on the specifics of the interactions between the different backbone amides and the ion. For example, not all sites have comparable effects on ion selectivity, sometimes not even in the same direction (i.e. in bCLC-k V166 ω and M427 μ decrease interanion discrimination while F426 ϕ renders the channel more selective for Cl⁻). In most cases we do see decreased interanion selectivity (Fig. 2, 4), and we explored the basis for such changes in M427 μ . In this case, our MD data suggest that the loss of selectivity is due, at least in part, to the degraded coordination of Cl⁻ in S_{ext} caused by the loss of the hydrogen bond at this mutation site. However, a precise mechanistic model would require an in-depth investigation of the effects at each position for every ion, which is beyond the scope of our manuscript. We now discuss this important point in Discussion on Page 14.

4. The dependence of the amide effects on the gating glutamate is not supported by the data in the current manuscript. For instance, the effects of backbone amide substitutions are qualitatively similar on the WT and V166E backgrounds in bCLC-k and on the WT and E166A backgrounds in CLC-0, with the exception of P_{NO3} in M427 μ for bCLC-k and Pbr of F418 ϕ in CLC-0. These results are therefore not consistent with a dependence on the gating glutamate, which might in fact be expected if the underlying mechanisms involve primarily partial charges as suggested by the amide-to-ester mutations. Please provide additional data and/or analysis to support this claim and the resulting interpretation in the discussion or modify the relevant texts accordingly.

We thank the reviewer for raising this important point. We agree that the correlation between the effects of substitutions at positions lining S_{cen} and S_{ext} and the presence of a glutamate at position 166 in CLC-0 and bCLC-k is imperfect. However, we do see a general trend where substitutions introduced in channels lacking E166 have larger effects than those introduced in channels with E166, indicating that Glu_{ex} modulates the roles of backbone amides in selectivity. We modified the text in Abstract (Pg. 2), Introduction (Pg. 4), Results (Pg. 9-10), and Discussion (Pg. 13), and revised Supplementary Fig. 6A-B accordingly.

5. The PMF profiles for Cl, Br, and NO₃ in bCLC-k overlap substantially across the selectivity filter, which were calculated for single-ion occupancy. The authors should explain how this is consistent with the >3-fold selectivity of Cl versus NO₃ and iodide and address the limitations of either the experimental or computational approaches and the potential role of multi-ion occupancy in anion selectivity. Please also include the data for iodide as it has the lowest relative permeability amongst the anions tested.

We thank the reviewer for the comment and suggestion. As requested, we performed additional PMF calculations for iodide, which shows consistent results with expectations and with PMF calculations for the other anions, namely I⁻ has strong interactions with the pore and presents a global minimum at S_{ext}. Results in Fig. 5, and Fig. 5-Supplement 2 have been updated accordingly. The PMF calculations here are performed to only probe the dominant interactions of the anion

with the protein environment (e.g., backbone amide) along the permeation pathway, specifically highlighting the strong interaction with selectivity filter. Given their single-ion occupancy nature, these calculations cannot provide full insight into the actual mechanism of ion permeation, which may explain the selectivity between the ions. We clearly point out this shortcoming in the manuscript (Pg. 12): “These results exemplify the critical contribution of backbone amide coordination for the anions within the pore. While this treatment does not represent the prevailing multi-ion permeation mechanism in these channels, it will probe the environment of the anions while in the pore and provide a reliable approximation for the energetics experienced by them.”

6. From the simulations, no clear evidence is presented on how disrupting the M427 amide can lead to an increase in the relative permeation of Br and NO₃ if amide-stabilized ion occupancy is a critical determinant for anion selectivity. The energy minima for Br are the most severely perturbed yet P_{Br} is increased in the experiments shown in Fig. 2B and the energy minima for NO₃ are similar to that of Cl yet P_{NO₃} is 2-fold less than that of Cl in Fig. 2B. The authors should also include the perturbation of ion occupancy for Br and NO₃ in Fig. 5C as Br binding is expected to be more substantially weakened by the mutation than Cl according to the PMF profiles shown in Fig. 5A and B.

As pointed out in response to the previous comment, the PMF calculations here are only to probe the protein environment felt by a permeating anion and are far from providing detailed mechanistic insight that would be needed to explain how ions are selected by the protein in a currently-unknown, but likely multi-ion, mechanism. Given the complexities of such mechanisms, which are not captured by the PMF calculations, the PMF profiles of the mutants can only be used to show the removal of the strong interactions with the selectivity filter upon the mutation of the backbone atoms.

To follow the second part of the reviewer’s comment, we now have added images demonstrating the perturbation of all the ions in both wildtype and mutant systems (revised Fig. 5C).

7. Given the spread of the data and that the changes are modest in most cases, the authors should explicitly include statistical comparison in Fig. 1-4. Please include the individual data points to help readers visualize the underlying spread of the data in Fig. 1-4.

We thank the reviewer for this suggestion. Initially we chose to mainly focus on the changes in rank order selectivity sequences. However, we agree it is important to rigorously establish the statistical significance of the measured changes. To this end we used Student’s t-test to determine the statistical significance of the effects for each mutant and for each ion. The statistical analysis, along with the scatter distribution of individual data points, is now reported in the new Supp. Fig. 5 and in the new Supp. Table 1 (where we also list the values for the permeability ratios in all conditions as mean±StDev). Overall, the statistical analysis supports the significance of the effects for all relevant constructs we mention in the text. We note that in our analysis we compared each α -hydroxy substitution to its parent construct. Thus, each single mutant was compared to the WT channel whereas each double mutant was compared to the corresponding E166A or V166E constructs. This is now stated in the methods section (Pg. 18) and in the legends to Supp Fig. 5 and Supp. Table 1. We prefer to keep this information in supplementary to keep main text figures easier to view and because the statistical analysis compares the relative permeability of each

mutant construct to that of the WT for each ion, so it is easier to identify the statistical significance of the effects by grouping the effects by ion rather than by mutant construct.

Reviewer #2 (Remarks to the Author):

This nice piece of work significantly advances our understanding of ClC channel permeation and selectivity.

Experiments are well-designed, the paper is very well written, and I have only minor criticism concerning data presentation:

We thank the reviewer for their positive assessment of our work.

1. If the same traces are shown in e.g. Fig 1E and Fig 1-Suppl. 2, it should be stated in the legend.

We apologize for the confusion. This is now stated explicitly in the Legend to Supp. Fig. 2.

2. It is better to show averaged traces and averaged I-V relationships rather than rather than obtained from a single cell (even though I do not doubt that they are indeed representative).

We thank the reviewer for this suggestion. However, one of the key purposes of Supplementary Fig. 3 is to illustrate the lack of drift in V_{rev} after perfusions to different anions, hence the use of un-normalized I-Vs and of showing 3 curves for Cl^- from a single cell. Additionally, since not all cells survived the full range of solution exchanges the numbers of replicates differ between different ions. Finally, because the order of Br^- and NO_3^- perfusions was at times inverted (see Methods), the conditions of the I-V for the second and third Cl^- I-V relationships are not constant. For these reasons we prefer to keep the representative I-Vs from a single cell. We now show individual data points and added a thorough statistical analysis of our results in the new Supp. Fig. 5 and Supplementary Table 1 to evaluate the significance of the observed changes as well as to illustrate the variability of individual measurements.

3. It would be nice to see all data points with bars for averages and error bars depicting SD rather than SEM for PX/PCl. This way the variability of measurements is not masked by the sample size.

We agree with the reviewer. We have now replotted all main text figures with SD rather than SEM. We added the individual data points in new Supplementary Fig. 5, together with the accompanying statistical analysis (please see our response to Reviewer 1 comment 7 above for a detailed discussion).

4. It may be nice to have WT values for reference in Fig2B,C and Fig4.

We apologize for the confusion; the dashed lines represent the WT values in these panels. To avoid confusion, we added the bars corresponding to the WT and reference mutants (E166A for CLC-0 and V166E for bCLC-k) where appropriate.

Other minor points:

1. Fig 2 legend: should be P_{NO_3} rather than PNO

2. P.15: The nonsense suppression method to site-specifically replace amino acids with pore-lining backbone amides with their α -hydroxy acid equivalents. Sentence has no verb.

Changed and changed, thank you!

Reviewer #3 (Remarks to the Author):

Overall, this is a well constructed study, and the use of unnatural amino acids to probe the role of backbone amides is well conceived and interesting.

We thank the reviewer for appreciating our work.

1. A defining structural difference between the bCLC-k and h-CLC-1 channels is the position of the central Serine side chain (Ser_{cen}), and this is discussed as a source for functional differences. However, it is possible that in physiological conditions this side chain could be dynamic and move between these positions and that the structures may only be capturing one state in their conditions (eg due to low temperature). Because this is the basis for much of the mechanistic differences proposed for the channels, I would like to see a discussion as to whether this is a robust difference between the two channels. Is there any evidence of side chain motion in the MD simulations of bCLC-k conducted here?

We thank the reviewer for bringing up this important point. It is possible that the C-D loop is flexible and can adopt multiple conformations. However, we see no evidence for two stable conformations of the C-D loop and Ser_{cen} from our simulations. If there are indeed two stable states, the transitions between them must happen on timescales longer than those explored in the simulations.

For the bar graphs of selectivity ratio in Fig 2 (B,C) and Fig 4 a statistical analysis to test whether the changes between the WT and mutants are statistically significant is needed. Because this is the key experimental data, it is important to be clear as to what changes are significant.

We agree with the reviewer this is an important point. Individual data points and statistical analysis are now shown in the new Supplementary Fig. 5. Please, see our response to Reviewer 1 point 7 above for a detailed discussion of our choices.

The MD simulations suggest that the external binding site, S_{ext}, has the strongest binding affinity. Can the changes in selectivity seen between the different channels *especially for Cl⁻ vs Br⁻) be explained by the presence or absence of this site? For example, if this site had the greatest selectivity, then removing it (eg M427u) could remove the Cl⁻ to Br⁻ selection. To better explore this, I would really like the selectivity of this site to be examined using the MD simulations with FEP or TI calculations. A clear result from the MD simulations as to the relative selectivity of different locations along the pore would make the origins of selection more clear.

We thank the reviewer for the suggestion. Following the suggestion, we have performed FEP simulations to estimate relative binding strengths of Cl⁻ and Br⁻ at the S_{ext} site for both WT and

mutant systems. FEP simulations were performed by making the ions disappear from the bulk solution and appear at the S_{ext} site, capturing their binding free energy from solution (ΔG). The relative binding free energies ($\Delta\Delta G$) were then obtained with respect to the lowest ΔG , which was for the case of Br^- binding to WT (set to 0). These values are reported in the table below.

Similar to PMF profiles, the FEP simulations at best only indicate that mutation at backbone amide reduces the binding strength of both ions at the selectivity region. The degree in which this will affect the overall permeation rate of the different ions can be significantly affected by the ion permeation mechanism, which likely differs from the single-ion treatment used here, as well as by other factors. As these results do not provide additional insight into the mechanism, beyond the PMF results, we have not included them in the revised manuscript.

Relative binding strengths ($\Delta\Delta G$) of Cl^- and Br^- at the S_{ext} site for both WT and M427 μ systems

Ion	System	$\Delta\Delta G$ (kcal/mol)	Statistical Error
Cl^-	WT	3.5	0.159
Br^-	WT	0.0	0.205
Cl^-	M427 μ	6.6	0.178
Br^-	M427 μ	13.7	0.141

I recommend plotting the ion hydration and protein coordination number as a function of position along the pore. This can be determined from the MD simulations used to make the PMFs and can help to understand the origin of the energy wells and the differences between the ions. The degree of ion dehydration is a well discussed mechanism for generating selection between cations, and has been shown to similarly play a role for anions (Richards et al Small, 8: 1701-1709, 2012; Phys Chem Chem Phys, 14: 11633-11638, 2012). This information could assist in clarifying the origins of selection in the current study.

We thank the reviewer for the suggestion. We have now added the coordination analysis of all ions for both WT and mutant systems to the revised Supplementary Fig. 8. Also, we have cited the suggested publications in the revised manuscript: “Furthermore, coordination analysis of the permeant anions suggests that the M427 μ mutation affects the hydration and ion-protein interaction patterns beyond the selectivity filter (Supplementary Fig. 8), which we believe could account for the PMF changes observed in other regions of the pore after the mutation. Previous MD studies have shown that hydration and protein coordination of ions along the pore can affect the energetics of the ion permeation process (Richards et al., 2012a; Richards et al., 2012b).”

I recommend being a bit more nuanced in comparing the role of the backbone amides, and the serine side chain in defining selectivity. It may be the case that both play some role, rather than being just one or the other.

We apologize for not clearly conveying our thoughts on this critical point. We do not think that backbone amides are the only determinants of anion selectivity. Indeed, in some CLCs the serine on the C-D loop plays a critical role in determining ion selectivity. Further, other side chains can also play roles as shown by Lagostena et al. (Lagostena et al., 2019) for CLC-Ka. Available structures together with our functional data suggest that backbone amides are structurally

conserved determinants of anion selectivity, but it is clear other elements also play a role in determining the selectivity of each individual CLC. We have therefore expanded our discussion in Pg. 12-13 to highlight the importance of multiple structural elements in anion selectivity.

Fig 2. In the legend, please explain the meaning of the dashed lines in panels B and C (I did work it out, but better to be explicit)

Following the suggestion of reviewer #2, we replaced the dashed lines with the bars for the WT constructs and state in the Figure 2 legend that the bars are the same as those reported in Fig. 1.

Fig 4. Include the WT values for reference

In Fig. 4 we evaluate the effects of α -hydroxy substitutions on a different background than the WT constructs used in Fig. 2, the E166A mutant for CLC-0 and the V166E for bCLC-k. Since these mutants themselves have small effects on ion selectivity (Fig. 4, Supplementary Fig. 5), we think it is more appropriate to use them as references rather than the WT constructs.

References

- Lagostena, L., G. Zifarelli, and A. Picollo. 2019. New Insights into the Mechanism of NO₃⁻ Selectivity in the Human Kidney Chloride Channel ClC-Ka and the CLC Protein Family. *Journal of the American Society of Nephrology*. 30:293-302.
- Lu, T., A.Y. Ting, J. Mainland, L.Y. Jan, P.G. Schultz, and J. Yang. 2001. Probing ion permeation and gating in a K⁺ channel with backbone mutations in the selectivity filter. *Nature Neuroscience*. 4:239-246.
- Park, E., and R. MacKinnon. 2018. Structure of the CLC-1 chloride channel from Homo sapiens. *eLife*. 7:e36629.
- Richards, L.A., A.I. Schäfer, B.S. Richards, and B. Corry. 2012a. The importance of dehydration in determining ion transport in narrow pores. *Small*. 8:1701-1709.
- Richards, L.A., A.I. Schäfer, B.S. Richards, and B. Corry. 2012b. Quantifying barriers to monovalent anion transport in narrow non-polar pores. *Physical Chemistry Chemical Physics*. 14:11633-11638.
- Valiyaveetil, F.I., M. Sekedat, R. MacKinnon, and T.W. Muir. 2006. Structural and Functional Consequences of an Amide-to-Ester Substitution in the Selectivity Filter of a Potassium Channel. *Journal of the American Chemical Society*. 128:11591-11599.
- Wang, K., S.S. Preisler, L. Zhang, Y. Cui, J.W. Missel, C. Grønberg, K. Gotfryd, E. Lindahl, M. Andersson, K. Calloe, P.F. Egea, D.A. Klaerke, M. Pusch, P.A. Pedersen, Z.H. Zhou, and P. Gourdon. 2019. Structure of the human ClC-1 chloride channel. *PLOS Biology*. 17:e3000218.

Reviewers' Comments:

Reviewer #1:

Remarks to the Author:

The authors have addressed most of my concerns. However, the manuscript will be strengthened by considering the following points.

1. A direct comparison between cation and anion channels might not be appropriate given that the coordination mechanisms are different (backbone carbonyl vs backbone N-H) and that the alpha-hydroxyl substitution replaces the N-H but not directly the carbonyl, please revise.

4. Please perform statistical analysis between the effect of the alpha-hydroxyl substitution on the wild-type versus the effect of the alpha-hydroxyl substitution on the mutant backgrounds to evaluate if the effect of the substitution is made more pronounced by the mutant background. I would suggest that the effect of the alpha-hydroxyl substitution on the wild-type versus the effect of the alpha-hydroxyl substitution on the mutant backgrounds are plotted on the same graph in order for the reader to make a direct comparison. The observed trends and hence conclusions should be based on statistical analysis for both bCIC-k and CIC-0, which should also be shown explicitly in the main figure (Fig. 4). If the analyses suggest otherwise, the conclusions should be revised.

7. Although the authors have provided more statistical analyses in the supplement, it is crucial that these analyses appear in the main figures to enable the reader to directly refer to the comparisons that the authors are trying to make. Please include them directly in the main figures, as they are essential to the conclusions that the authors wish to draw.

Once these points are fully addressed as suggested, the manuscript would be an informative contribution.

Reviewer #2:

Remarks to the Author:

The authors have satisfactorily addressed my points, but a small point remains:

The figures in the main text do not seem replotted: the error bars are exactly as they were before, but now the legends state that they represent St.Dev. What do they actually represent? Was it a mistake in the original version, as it now seems while comparing the SF5 to the main text? It would be nice to have all data points visible not only in the supplementary, but also in the main figures.

After addressing this small point, this interesting manuscript seems ready for publication.

Reviewer #3:

Remarks to the Author:

The authors have taken all the comments from the reviewers seriously and made a very large effort to respond and incorporate the suggestions into the manuscript. I believe they have done an excellent job and am satisfied that the revised manuscript makes an important contribution to the field and that it is technically sound.

My only tiny comment would be to include a sentence conveying the response to my first concern in the main text. The following statement from the response to reviews could be included in the discussion:

It is possible that the C-D loop is flexible and can adopt multiple conformations. However, we see no evidence for two stable conformations of the C-D loop and Sercen from our simulations. If there are indeed two stable states, the transitions between them must happen on timescales longer than those explored in the simulations.

We thank the reviewers for their positive assessment of our work. Below we describe how we addressed their comments (in blue).

Reviewer #1 (Remarks to the Author):

The authors have addressed most of my concerns. However, the manuscript will be strengthened by considering the following points.

1. A direct comparison between cation and anion channels might not be appropriate given that the coordination mechanisms are different (backbone carbonyl vs backbone N-H) and that the alpha-hydroxyl substitution replaces the N-H but not directly the carbonyl, please revise.

We thank the reviewer for raising this point. We edited our text in the abstract to clarify that “We propose that backbone-ion interactions are determinants of Cl⁻ specificity in CLC channels in a mechanism reminiscent of that described for K⁺ channels.” and in the discussion to “Notably, equivalent manipulations do not affect the selectivity sequence of K⁺ channels 63, 64, supporting the idea that interactions between backbone amides and permeating anions are specifically important for selectivity in CLCs.” The similarity in the two mechanisms lies in the role of the interactions between backbone elements and the permeating ions, and the comparison with the lack of the effects of backbone amide substitutions in K⁺ channels supports the idea that the effects we see here are specific, rather than resulting from generic alterations in protein structure due to the backbone substitutions.

4. Please perform statistical analysis between the effect of the alpha-hydroxyl substitution on the wild-type versus the effect of the alpha-hydroxyl substitution on the mutant backgrounds to evaluate if the effect of the substitution is made more pronounced by the mutant background. I would suggest that the effect of the alpha-hydroxyl substitution on the wild-type versus the effect of the alpha-hydroxyl substitution on the mutant backgrounds are plotted on the same graph in order for the reader to make a direct comparison. The observed trends and hence conclusions should be based on statistical analysis for both bCLC-k and CLC-0, which should also be shown explicitly in the main figure (Fig. 4). If the analyses suggest otherwise, the conclusions should be revised.

We thank the reviewer for these helpful suggestions. We changed the main text figures to show individual data points and indicate the statistical significance of the effects, additionally we revised our statistical analysis to include a Bonferroni correction to our significance threshold. We do not think a statistical comparison between the effects of the alpha-hydroxyl substitutions on the background of WT and Glu_{ex} mutant channels is appropriate because the mutants affect (slightly) channel selectivity. Thus, we believe that limiting the statistical comparisons to the parent constructs is the most appropriate course of action.

7. Although the authors have provided more statistical analyses in the supplement, it is crucial that these analyses appear in the main figures to enable the reader to directly refer to the comparisons that the authors are trying to make. Please include them directly in the main figures, as they are essential to the conclusions that the authors wish to draw.

Done.

Reviewer #2 (Remarks to the Author):

The authors have satisfactorily addressed my points, but a small point remains:

The figures in the main text do not seem replotted: the error bars are exactly as they were before, but now the legends state that they represent St.Dev. What do they actually represent? Was it a mistake in the original version, as it now seems while comparing the SF5 to the main text? It would be nice to have all data points visible not only in the supplementary, but also in the main figures.

We apologize for the confusion. It is possible that in some of our original graphs we had reported the St Dev while indicated that they represented the St. Error. We have now checked that all error bars represent the St. Dev. of the data and added individual data points to the main text figures.

After addressing this small point, this interesting manuscript seems ready for publication.

Reviewer #3 (Remarks to the Author):

The authors have taken all the comments from the reviewers seriously and made a very large effort to respond and incorporate the suggestions into the manuscript. I believe they have done an excellent job and am satisfied that the revised manuscript makes an important contribution to the field and that it is technically sound.

My only tiny comment would be to include a sentence conveying the response to my first concern in the main text. The following statement from the response to reviews could be included in the discussion:

It is possible that the C-D loop is flexible and can adopt multiple conformations. However, we see no evidence for two stable conformations of the C-D loop and Ser_{cen} from our simulations. If there are indeed two stable states, the transitions between them must happen on timescales longer than those explored in the simulations.

We thank the reviewer for the helpful suggestion. We added the following sentence to the discussion: “The differential role of the Ser_{cen} side chain in anion selectivity of different CLC homologues could reflect the flexibility of the C-D loop to interchange between conformations where Ser_{cen} points towards or away from the S_{cen} site (Fig. 1). However, in our bCLC-k simulations we see no evidence for two stable conformations of this loop, suggesting that if there are indeed two stable states the transitions between them must happen on timescales longer than those explored in the simulations.”